# Overexpression of Sweet Potato Carotenoid Cleavage Dioxygenase 4 (*IbCCD4*) Decreased Salt Tolerance in *Arabidopsis thaliana*

**DOI:** 10.3390/ijms23179963

**Published:** 2022-09-01

**Authors:** Jie Zhang, Liheng He, Jingjing Dong, Cailiang Zhao, Ruimin Tang, Xiaoyun Jia

**Affiliations:** 1College of Agriculture, Shanxi Agricultural University, Taigu 030801, China; 2College of Life Sciences, Shanxi Agricultural University, Taigu 030801, China

**Keywords:** *IbCCD4*, salt tolerance, carotenoid cleavage dioxygenase, apocarotenoids

## Abstract

Salt stress has a serious impact on normal plant growth and yield. Carotenoid cleavage dioxygenase (CCD) degrades carotenoids to produce apocarotenoids, which are involved in plant responses to biotic and abiotic stresses. This study shows that the expression of sweet potato *IbCCD4* was significantly induced by salt and dehydration stress. The heterologous expression of *IbCCD4* in Arabidopsis was induced to confirm its salt tolerance. Under 200 mM NaCl treatment, compared to wild-type plants, the rosette leaves of *IbCCD4*-overexpressing *Arabidopsis* showed increased anthocyanins and carotenoid contents, an increased expression of most genes in the carotenoid metabolic pathway, and increased malondialdehyde (MDA) levels. *IbCCD4*-overexpressing lines also showed a decreased expression of resistance-related genes and a lower activity of three antioxidant enzymes: peroxidase (POD), superoxide dismutase (SOD), and catalase (CAT). These results indicate that *IbCCD4* reduced salt tolerance in *Arabidopsis*, which contributes to the understanding of the role of IbCCD4 in salt stress.

## 1. Introduction

Salt stress seriously affects the growth and development of plants [1]. Soil salinity is among the global challenges that seriously affect agricultural productivity [2]. Globally, the area of salinized soil is 935 million ha (M ha) [3], and >70% of the total area is located in arid and semi-arid regions of India, Pakistan, Australia, African countries, China, the Middle East, and the United States [4]. Moreover, the area of soil salinity is expanding at a rate of 1–2 M ha per year [5]. It is urgent to develop crops with high levels of salt tolerance.

Carotenoids, a general term for a class of important natural pigments, belong to isoprenoid compounds, which bring yellow, orange, and red color to plants. More than 700 kinds of carotenoids have been found to be ubiquitous in animals, higher plants, fungi, algae, and bacteria [6,7]. These natural pigments have various biological functions in plants [7,8,9]. For example, their participation in photosynthesis as a photosynthetic pigment can also prevent the plants from damage by photooxidation [10,11], they give bright colors to petals and fruits, which can attract insects and animals to pollinate and transmit seeds [12], and they are involved in responding to abiotic stresses [13,14].

Carotenoid accumulation in plants depends on the metabolic balance between carotenoid biosynthesis and degradation. The oxidative cleavage of carotenoids in higher plants is mainly catalyzed by carotenoid cleavage dioxygenase (CCD). CCD is a kind of non-heme iron enzyme which contains the retinal pigment epithelial membrane protein (RPE65) domain and is responsible for binding Fe^2+^ [15]. It can be divided into carotenoid cleavage dioxygenases (CCDs) and 9-*cis* epoxy carotenoid dioxygenases (NCEDs) according to different substrates. A total of nine CCD members were identified in *Arabidopsis thaliana*, including four CCDs (AtCCD1, AtCCD4, AtCCD7, and AtCCD8) and five NCEDs (AtNCED2, AtNCED3, AtNCED5, AtNCED6, and AtNCED9) [16].

Studies have shown that CCD subfamily genes play important roles in the growth and development of plants. *CCD*1s are mainly involved in the formation of flavor volatiles in different species [17,18,19]. *CCD*4s are involved in the white color formation of flowers, fruits, and tubers in plants [20,21,22,23,24,25]. *CCD*7 and *CCD*8 are mainly involved in the formation of strigolactones (SLs) to regulate the morphogenesis of plant shoots and roots [26,27]. It has also been reported that CCDs respond to biotic and abiotic stresses [27,28,29]. The tomato mutant *ccd8* is more sensitive to pathogens (*Botrytis cinerea* and *Alternaria alternata*), indicating that *SlCCD*8 is involved in the plant defense against pathogens [30]. *BoCCD*1 and *BoCCD*4 are involved in the responses to drought and salt stress in *Brassica oleracea* [31]. The overexpression of *CsCCD4b* from *Crocus sativus* was shown to improve the response to salinity in transgenic Arabidopsis plants [32]. However, there are fewer studies on the functional characterization of CCDs of sweet potato under abiotic stress.

Sweet potato (*Ipomoea batatas* L.) is an important food, feed, and industrial raw material, and it occupies an important position in the world food production security [33]. In particular, the orange-fleshed sweet potato enriched in carotenoids is a cheap source of dietary carotenoids. Carotenoids have benefits for human health, with many physiological functions including anti-inflammatory, anti-mutagenic, anti-cancer, and anti-oxidative properties [34,35,36,37,38]. In recent years, with the increase in the area of soil salinization, high salt stress has become one of the adverse environmental factors seriously affecting the yield increase of sweet potato [39]. Thus, developing salt-resistant sweet potato varieties is important to improve quality and yield performance. Some sweet potato varieties with salt-resistance (such as LM79, Taizhong-9, Shushu-7, Longshu-1, and Xushu-18) have been obtained by classical breeding [40,41,42]; however, molecular breeding is faster and more efficient [43]. The functional analysis of salt stress-related genes can lay a solid foundation for breeding salt-resistant varieties and improving the yield of sweet potato under salt stress. In this study, we cloned the genomic sequence of *IbCCD4* and analyzed its function in response to salt stress in *Arabidopsis thaliana*, aiming to provide a reference base for breeding new sweet potato varieties with improved salt tolerance by using gene editing technology targeting the genomic sequence of *IbCCD4*.

## 2. Results

### 2.1. Bioinformatics Analysis of IbCCD4

The ORF encoding protein of IbCCD4 contains a conserved domain of RPE65 (Figure 1A). In this study, we cloned the full-length genomic sequence of *IbCCD*4 from the tuberous root of XuShu18. The genomic DNA of *IbCCD*4 was 2993 bp, with an intron (Figure 1B). The multiple sequence alignment of CCD4 proteins from 16 plant species with high sequence similarity (Appendix A) showed that they contained four H residues and three G and D residues. The IbCCD4 protein sequence had the highest similarity with that of *Ipomoea triloba* (XP_031124907.1), at 95.62%. The similarity with *Ipomoea nil* (XP_019156361.1), *Crocus sativus* (ACD62477), potato (*Solanum tuberosum*, XP_006359966.1), tomato (*Solanum lycopersicum*, XP_004246004.1), grape (*Vitis vinifera*, AGT63321.1), apple (*Malus domestica* XP_008340019.2), *Osmanthus fragrans* (ABY60887.1), *chrysanthemum* (*Dendranthema morifolium*, BAF36656.2), peach (*Prunus persica*, PRUPE_1G255500), *Arabidopsis thaliana* (AT4G19170), *Zea mays* (PWZ28009.1)*,* wheat *(Triticum aestivum* QEX50885.1), *Citrus* (CitCCD4, AB781691), and *Citrus clementina* (CitCCD4b, ABC26012) was 91.26%, 77.41%, 71.26%, 70.50%, 70.48%, 70.32%, 69.79%, 67.85%, 66.01%, 65.38%, 52.50%, 54.59%, 47.17%, and 47.00%, respectively.

A phylogenetic tree shows the evolutionary relationship between IbCCD4 and CCD4 proteins is derived from 15 other plant species, indicating that IbCCD4 has the closest genetic relationship with its diploid ancestor species, ItCCD4 (Figure 1C).

As showed in Figure 1C, these proteins shared a majority of motifs because they are members of the CCD4 group. Consistent with the phylogenetic tree, closely related proteins contain almost similar motif numbers and compositions. IbCCD4 shares the same motif number and composition with InCCD4 and ItCCD4, due to them being members of Ipomoea, and similar results were observed between StCCD4 and SlCCD4 and CitCCD4 and CitCCD4b, respectively. On the other hand, the number and composition of motifs differed among proteins that were distantly related, such as IbCCD4 and CsCCD4. IbCCD4 contains motifs 20, 23, 17, 12, 19, 22, and 21, but these motifs are absent in CsCCD4b. Motif 23 is absent in most of the CCD4 proteins analyzed in this study, excluding IbCCD4, InCCD4, ItCCD4, and PpCCD4. Compared with IbCCD4, motifs 22 and 23 are absent in MdCCD4 and AtCCD4, motif 23 is absent in OfCCD4, and motifs 20, 23, 17, 15, 19, and 22 are absent in CitCCD4 and CitCCD4b.

### 2.2. IbCCD4 Response to Salt and Dehydration Stress

The dynamic expression of the *IbCCD*4 gene was induced in leaves under salt and dehydration stress and was assessed at the following time points: 0, 6, 12, 24, 48, and 72 h. As shown in Figure 2, the expression of *IbCCD4* was significantly induced under salt and dehydration stresses (*p* < 0.05). The transcript level of *IbCCD4* was also strongly induced by 250 mM NaCl of stress treatment. These results suggest an *IbCCD4* response to salt and dehydration stress.

### 2.3. The Ectopic Expression of IbCCD4 Reduced Salt Tolerance in Transgenic Arabidopsis

After confirmation by PCR and qRT-PCR, three T_3_ homozygous lines (L4, L5, and L6) overexpressing *IbCCD4* were selected for the salt stress assay (Figure 3).

After seven days of germination on 1/2 MS medium, the primary root length of seven-day-old *IbCCD4*-overexpressed seedlings was longer than that of control seedlings (Figure 4A, 0 d). It was found that L4, L5, and L6 had shorter roots than WT after 10 days of growth under 0 mM NaCl and fewer lateral roots than WT (Figure 4B,C). After 10 days of growth under 125 mM NaCl, the root growth of WT and L4, L5, and L6 was inhibited, and the primary root length and the number of lateral roots of L4, L5, and L6 were significantly lower than those of WT (Figure 4B,C). After 10 days of growth under 150 mM NaCl, the primary root length of WT and L4, L5, and L6 was significantly inhibited, and L4, L5, and L6 had fewer lateral roots than WT (Figure. 4A).

The seven-day-old seedlings of the wild-type plants (WT) and *IbCCD*4-overexpressing T_3_ lines (L4, L5, L6) were transplanted into soil, and after normal growth of 15 days, they were irrigated with 200 mM NaCl for salt stress treatment. Before treatment, there was no significant difference in the growth status and anthocyanin content in the rosette leaves among L4, L5, L6, and WT (Figure 5A, 0 d). After 7 days of salt stress treatment, the rosette leaves of the L4, L5, and L6 lines showed obvious purple, and their anthocyanin content was significantly higher than that of WT (Figure 5B).

### 2.4. The Ectopic Expression of IbCCD4 Increased the Total Carotenoid Content and the Expression Levels of Most Carotenoid Biosynthesis Genes in Transgenic Arabidopsis

After treatment with 200 mM NaCl for 7 days, the total carotenoids and the expression of carotenoid biosynthesis genes in the rosette leaves of wild-type (WT) and transgenic lines (L4, L5, L6) were determined (Figure 6). It was found that the total carotenoid content in the L4, L5, and L6 lines was significantly higher than that in WT (*p* < 0.05) (Figure 6A). The qRT-PCR analysis showed that the expression levels of *AtPSY, AtPDS, AtCHYB, AtLCYE,* and *AtZEP* of L4, L5, and L6 were higher than those of WT. Notably, the expression of *AtCHYE* was higher in L4 and lower in L5 than that of WT, and there was no significant difference between L6 and WT. The expression level of *AtLCYB* in L4 was higher than that of WT, while there was no significant difference among WT, L5, and L6 (Figure 6B).

### 2.5. The Ectopic Expression of IbCCD4 Decreased the Expression of Resistance-Related Genes in Transgenic Arabidopsis under Salt Stress

After treatment with 200 mM NaCl for 7 days, the expression of resistance-related genes was analyzed in the leaves of WT and the transgenic lines (L4, L5, and L6) (Figure 7). The expression of *AtCCD4* and *IbCCD4* decreased in WT, L4, L5, and L6 under salt stress. The salt stress response genes *AtRD*29*a* and *AtKIN*2 were highly expressed under salt stress treatment, especially in WT. Under 200 mM NaCl stress, the expression of *AtSOD* in WT, L4, L5, and L6 increased, and the increased level of *AtSOD* in WT was significantly higher than that in L4, L5, and L6. It is notable that, under 200 mM NaCl salt treatment, the expression of *AtPOD* decreased in L4, L5, and L6 by 99.4%, 98.8%, and 99.8%, respectively, compared to the control, and by only 74% in WT.

### 2.6. The Ectopic Expression of IbCCD4 Decreased Antioxidant Enzyme Activity in Transgenic Arabidopsis under Salt Stress

The results of determining the activity for three antioxidants (Figure 8A–C) showed that, under normal growth conditions (Control), there was no significant difference in POD and CAT activity among L4, L5, L6, and WT, and the SOD activity in L4 and L5 was slightly higher than that in WT and L6. Under 200 mM NaCl stress, the SOD, POD, and CAT activity in WT, L4, L5, and L6 significantly increased, and the increased level in WT was significantly higher than that in L4, L5, and L6. Compared to the control, the CAT activity in WT, L4, L5, and L6 increased by 4.49, 0.21, 1.00, and 1.41 times, respectively, under 200 mM NaCl stress; the POD activity in WT, L4, L5, and L6 increased by 2.83, 1.68, 1.79, and 1.52 times, respectively; and the SOD activity in WT, L4, L5, and L6 increased by 3.16, 2.53, 2.62, and 2.29 times, respectively.

Moreover, there was no significant difference in MDA content in L4, L5, L6, and WT under normal growth conditions (control). Under 200 mM NaCl stress, the MDA content of WT, L4, L5, and L6 was significantly increased, but the MDA contents of L4, L5, and L6 were significantly higher than those of WT. Compared to the control, the MDA content in WT, L4, L5, and L6 increased by 4.75, 5.61, 6.19, and 7.14 times, respectively (Figure 8D).

## 3. Discussion

Previous studies have identified that *CCD4* genes from different plants responded to various stresses. The expression of *MdCCD4c* was shown to be upregulated under ABA and salt stress [29], and the expression of *OfCCD*4 was upregulated after ABA treatment [44]. *CsCCD4b* from *Crocus sativus* responded to dehydration, salt, and oxidative stresses [32]. In this study, the expression of *IbCCD4* in the leaves of sweet potato showed an increase followed by a decrease under salt stress and dehydration stress, indicating that it responded to these stresses.

It has been reported that carotenoid derivatives play a role in root growth. BYPASS was involved in production of a carotenoid derivative which negatively regulated root growth [45]. However, another report confirmed that an uncharacterized carotenoid derivative was required for root growth [46]. These results indicated that these carotenoid derivatives played different functions in root growth and development. In our study, the primary root length of *IbCCD4*-overexpressing Arabidopsis plants was shorter than that of WT (Figure 4A). This may be a result of some carotenoid derivatives, the cleavage products of IbCCD4, which may function as a negative regulator of root growth.

Anthocyanins are a class of water-soluble pigments. Anthocyanins accumulation can bring a purple color to leaves. Anthocyanin accumulation is considered to be a visible biomarker of plants that have suffered from abiotic or biotic stresses [47]. Many studies have shown that the leaves of the plants exhibit a purple color under abiotic stress [47,48,49], which is related to the accumulation of anthocyanins, which improve stress resistance. The accumulation of anthocyanins was increased under phosphate (Pi) starvation in Arabidopsis [48]. Anthocyanins accumulate in response to abiotic stress, such as high visible light levels, low temperature, or low phosphate [49], and the leaves of plants with weaker resistance will show purple earlier. In this study, there was no significant difference in the anthocyanin content of rosette leaves between transgenic lines (L4, L5, and L6) and WT under normal conditions (Figure 5, 0 d); however, after 7 days of 200 mM NaCl treatment, the rosette leaves of the transgenic lines L4, L5, and L6 showed obvious purple, in which higher anthocyanin levels were detected than those in WT (Figure 5B). These results indicate that the *IbCCD4*-overexpressing transgenic Arabidopsis is more sensitive to salt stress than WT.

The accumulation of carotenoids was regulated not only by carotenoid biosynthesis genes but also by carotenoid degradation genes [50,51,52]. After seven days of treatment with 200 mM NaCl, the total carotenoid content in rosette leaves of *IbCCD*4-overexpressing Arabidopsis was higher than that of WT (Figure 6A). The reason for the higher carotenoid content may be the higher expression of carotenoid biosynthesis genes, such as *AtPSY, AtPDS, AtCHYB, AtLCYE,* and *AtZEP* (Figure 6), and the decreased expression of carotenoid degradation genes, such as *AtCCD4* and *IbCCD4,* compared to WT (Figure 7).

Under abiotic stress, a large number of reactive oxygen species (ROS) are produced in plants, which can be eliminated by antioxidant enzymes (such as SOD, CAT, and POD). In this study, the expression of the salt stress responsive genes *AtRD29*a and *AtKIN2*, along with the ROS scavenging system gene *AtSOD*, was significantly increased after 7 days of 200 mM NaCl treatment, and the expression of *AtSOD* was higher in WT than that in *IbCCD*4-overexpressing lines (Figure 6). The SOD, POD, and CAT activity in WT, L4, L5, and L6 significantly increased after 7 days of 200 mM NaCl treatment, and the increased level in WT was significantly higher than that in L4, L5, and L6 (Figure 8A–C). MDA accumulation under stress will destroy the integrity of the cell membrane. The higher the content of MDA, the weaker the resistance of the plant. In this study, the MDA content in plants increased under 200 mM NaCl stress, and the increased level was higher in L4, L5, and L6 than that in WT (Figure 8D). This result may be due to the increased activity of CAT, POD, and SOD in WT (Figure 8).

Homologous CCD4s from different species used different carotenoid substrates to produce apocarotenoids. CsCCD4 from *Camellia sinensis* mainly cleaves β-carotene to produce β-ionone [53]. In citrus, β-cryptoxanthin and zeaxanthin are the main substrates of CitCCD4, forming β-citraurin [54], and CitCCD4b can cleave β-carotene to produce β-cyclocitral [55]. DcCCD4 from *Daucus carota* L. cleaves α- and β-carotene to form α- and β-ionone [23]. An *IbCCD4* gene (KM973214) from sweet potato was identified by Park et al. [56], and the sequence similarity showed 96.8% identity with the one in the present study (OM674440). They confirmed that IbCCD4 (KM973214) mainly cleaves β-carotene; however, the cleavage product has not been confirmed. Many studies have reported that β-ionone and β-cyclocitral are signal molecules involved in stress responses [57,58,59]. The overexpression of *CsCCD4b* from *Crocus sativus* improved the response to salinity in transgenic Arabidopsis plants [32]. In this study, the results indicate that the overexpression of *IbCCD4* decreased the salt tolerance of transgenic Arabidopsis plants. The inconsistency of these results suggests that different mechanisms of CCD4 are involved in stress responses. As showed in Figure 1C, these proteins share a majority of motifs because they are members of the CCD4 group. Consistent with the phylogenetic tree, the number and composition of motifs differed among proteins that are distantly related, such as IbCCD4 and CsCCD4, which may lead to different substrates and cleavage products. Closely related proteins, such as CitCCD4 and CitCCD4b, contain almost similar motif numbers and compositions (Figure 1C); however, their substrate and cleavage products are very different [43,44]. This suggests that, even if they contain the same motif, the substrates and cleavage products of CCD4 proteins may be different. The functional differentiation of CCD4 may occur in different plants, and the differences in their substrates and cleavage sites may result in different cleavage products that play different roles in stress responses. The degradation of carotenoids by IbCCD4 may produce some unknown cleavage products involved in salt stress responses in *Arabidopsis* plants. Further studies are needed to identify the cleavage products of IbCCD4.

## 4. Materials and Methods

### 4.1. Plant Materials

The sweet potato cultivars ‘Xushu18’ (XS18) and ‘Xuzishu3’ (XZS3) were provided by the Sweet Potato Research Institute, Chinese Academy of Agricultural Sciences (Xuzhou, China).

### 4.2. IbCCD4 Multiple Sequence Alignment, Phylogenetic Tree Construction, and Motif Analysis

According to the ORF sequence of IbCCD4 (GenBank accession number OM674440), which was previously identified from the tuberous root of XS18 in our laboratory, similar sequences from 15 other plant species were downloaded from the NCBI website (http://www.ncbi.nlm.nih.gov/, accessed on 1 August 2022). Multiple sequence alignment was carried out by DNAMAN software, and then the neighbor-joining method was used to construct a phylogenetic tree by MEGA7.0 with 1000 bootstrap replications. The conserved motifs in the CCD protein sequences were identified by the Multiple Expectation Maximization for Motif Elicitation (MEME) online software (http://meme-suite.org, accessed on 1 August 2022) using the following parameters: the optimum motif width was set from 6 to 50, and the maximum number of motifs was 25.

### 4.3. Salt and Dehydration Stress Treatment

Pieces of vine approximately 10 cm in size were cut from XZS-3 sweet potato plants, which were grown in the field for 30 d, and then placed in 1/2 Hoagland nutrient solution in an incubator (25 °C, 16 h light at 22 °C, 8 h dark). After 2 weeks of growth, to induce salt and dehydration stress, the cuttings were placed in 1/2 Hoagland nutrient solution with 250 mM NaCl or 30% polyethylene glycol (PEG 6000). The leaves were sampled at 0, 6, 12, 24, 48, and 72 h and frozen in liquid nitrogen quickly; then, they were stored at –80 °C.

### 4.4. Total RNA Extraction, cDNA Synthesis, and Expression Analysis by qRT-PCR

RNA was extracted using an RNA extraction kit (TaKaRa Biotechnology, Dalian, Liaoning, China). The RNA integrity was evaluated by 1% agarose gel electrophoresis. The RNA concentration and quality were measured by NanoDrop 2000C (Thermo Scientific, Waltham, MA, USA). First-strand cDNA was synthesized from 1 μg of total RNA according to the instructions of the PrimeScript^TM^ RT Reagent Kit with the gDNA Eraser (TaKaRa Biotechnology, Dalian, Liaoning, China). The primers for quantitative real-time fluorescence PCR (qRT-PCR) were designed using online NCBI tools (https://www.ncbi.nlm.nih.gov/tools/primer-blast/, accessed on 6 March 2021) (Appendix A). All RT-qPCR experiments were performed on the CFX96 PCR System (Bio-Rad, Hercules, CA, USA) using SYBR® Premix Ex Taq^TM^ (TAKARA Biotechnology, Dalian, Liaoning, China). The 2 ^−∆∆Ct^ method [60] was used to calculate the relative expression of genes, using *IbActin* as an internal reference.

### 4.5. Cloning of the IbCCD4 Genomic Sequence and Gene Structure Analysis

The ORF sequence of *IbCCD*4 (GenBank accession number OM674440) was used to predict the gene structure of the genomic DNA sequence of sweet potato, ‘Taizhong 6’, from the genome database (https://121.36.193.159/blast.html, accessed on 1 August 2022). Then, the DNA from ‘XS18’ was used as a template, and two primer pairs (*CCD*4-F-*KpnI* and *CCD*4-R and *CCD*4-F and *CCD*4-R-*XbaI*) were used for PCR amplification. After sequencing, the genomic sequence of *IbCCD*4 was obtained by splicing. Finally, the gene structure of *IbCCD*4 was obtained on the Gene Structure Display Server website (http://gsds.gao-lab.org/, accessed on 1 August 2022).

### 4.6. Generation of Transgenic Arabidopsis

The ORF sequence of *IbCCD4* without a stop code sequence was ligated to the expression vector pCambia1300. The expression of *IbCCD4* was driven by the CAMV35S promoter. The recombinant plasmid pCambia1300-*IbCCD4* was confirmed by restriction digestion and sequencing; then, it was transformed into the *Agrobacterium tumefaciens* strain GV3101. Arabidopsis transformation was conducted by the floral dip method [61], and the seeds of T_0_ generation were harvested. The transformants were screened on 1/2 MS medium containing 10 mg/L hygromycin until theT_3_ homozygous lines were obtained. PCR was conducted to confirm the positive transgenic plants using the specific primers (35S-F; CCD4-R-*Xba*I). Three T_3_ lines with a high expression of *IbCCD4* were selected for further analysis.

### 4.7. Salt Stress Assay of IbCCD4-Overexpressing Arabidopsis

For the NaCl stress assay on the plate, sterilized seeds of WT and *IbCCD4-*overexpressing T3 lines were germinated on 1/2 MS medium for 7 days in a growth chamber (23 °C day/20 °C night, 16 h light/8 h dark cycle). Seven-day-old seedlings were transferred to 1/2 MS medium containing 0 (control), 125, and 150 mM NaCl for salt stress, and the seedlings were scored for the primary root length at 0, 7, and 10 days after salt stress and the number of lateral roots at 10 days after salt stress.

For the stress assay of irrigation with 200 mM NaCl, the 7-day-old seedlings grown on the 1/2MS medium of WT and the T3 lines grown on the 1/2 MS medium were transferred to pots filled with a mixture of soil and vermiculite (1:3, *v*/*v*) for 15 days; then, they irrigated with 200 mM NaCl every 3 days. On the 7th day of irrigation, rosette leaves were sampled to determine the anthocyanin and carotenoid contents, the enzyme activity of the superoxide dismutase (SOD), peroxide (POD), catalase (CAT), and malonaldehydic acid (MDA) contents, and the expression of carotenoid metabolic-related genes and stress-related genes. The total carotenoid and anthocyanin contents in the leaves were detected using previously reported methods [62,63]. The MDA content and SOD, POD, and CAT activities were measured by a reagent according to the instructions (Nanjing Jiancheng Biotechnology Company, Nanjing, China). The SOD activity was determined by the hydroxylamine method (A001-1-2), the POD activity was determined by the colorimetric method (A084-3-1), the CAT activity was determined by the ammonium molybdate method (A007-1-1), and the MDA content was determined by the TBA method (A003-1-2). The expression levels of carotenoid metabolic-related and stress-related genes were analyzed by qRT-PCR using the specific primers (Appendix A). These experiments were conducted in triplicate.

### 4.8. Statistical Analysis of Data

All data were analyzed with SPSS software (Chicago, IL, USA, version 8.0) using Student’s *t*-test, and statistical significance was determined at *p <* 0.05. All figures were completed by Origin software (Northampton, MA, USA, version 2019).

## 5. Conclusions

In this study, the expression of *IbCCD4* was significantly induced under salt and drought stresses. The overexpression of *IbCCD4* reduced the salt tolerance of transgenic Arabidopsis. These results confirm the role of IbCCD4 in salt stress and provide a candidate gene for the development of new sweet potato varieties with improved salt tolerance by reducing the expression of *IbCCD4* through gene editing technology, such as CRISPR/Cas9.

## Figures and Tables

**Figure 1 ijms-23-09963-f001:**
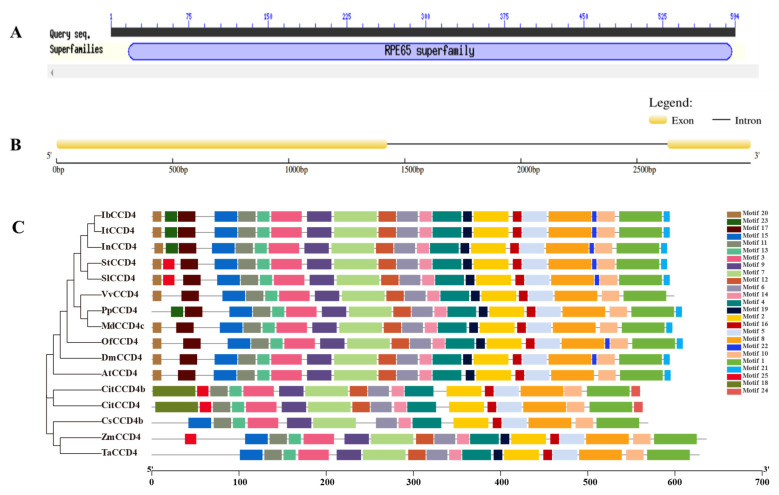
Bioinformatics analysis of *IbCCD4*: (**A**) conserved domain of IbCCD4; (**B**) gene structural analysis of *IbCCD4*; (**C**) phylogenetic tree and conserved motifs of the CCD4 proteins from IbCCD4 and 15 other plant species. The phylogenetic tree is shown in the left panel. GenBank accession numbers of the CCD4 proteins include IbCCD4 (*Ipomoea batatas,* ), ItCCD4 (*Ipomoea triloba*, XP_031124907.1), InCCD4 (*Ipomoea nil*, XP_019156361.1), StCCD4 (*Solanum tuberosum*, XP_006359966.1), SlCCD4 (*Solanum lycopersicum*, XP_004246004.1), VvCCD4 (*Vitis vinifera*, AGT63321.1), OfCCD4 (*Osmanthus fragrans*, ABY60887.1), DmCCD4 (*Dendranthema morifolium,* BAF36656.2), PpCCD4 (*Prunus persica*, PRUPE_1G255500), AtCCD4 (*Arabidopsis thaliana*, AT4G19170), MdCCD4c (*Malus domestica,* XP_008340019.2), ZmCCD4 (*Zea mays,* PWZ28009.1)*,* CitCCD4 (*Citrus*, AB781691), CitCCD4b (*Citrus clementina,* ABC26012), CsCCD4b (*Crocus sativus*, ACD62477), and TaCCD4 *(Triticum aestivum,* QEX50885.1). Scale bar indicates the nucleotide substitution per site. In the right panel, colored boxes represent different motifs.

**Figure 2 ijms-23-09963-f002:**
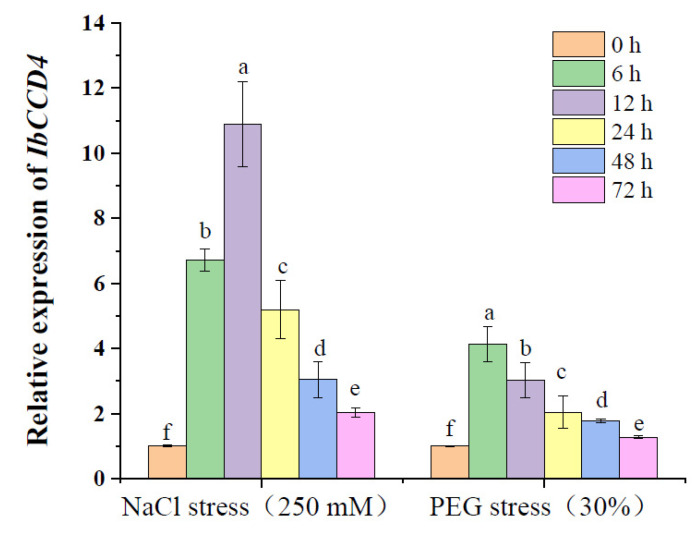
The expression of *IbCCD4* in leaves of sweet potato under 250 mM NaCl of salt stress and 30% PEG of dehydration stress. Different letters indicate the significant differences under the same treatment (*p* < 0.05).

**Figure 3 ijms-23-09963-f003:**
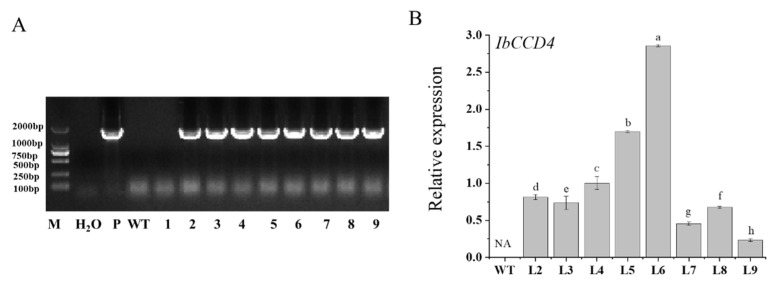
Confirmation of *IbCCD4*-overexpressing lines of Arabidopsis by PCR and qRT-PCR. (**A**) Selection of *IbCCD4*-overexpressing T_0_ lines by PCR. M: marker; P: the vector hosting the *IbCCD4* gene; WT: the non-transgenic Arabidopsis (Col); 1–9: the samples from the *IbCCD4*-overexpressing T_0_ lines; (**B**) the relative expression of *IbCCD4* in T_3_ transgenic lines. Different letters indicate the significant differences under the same treatment (*p* < 0.05).

**Figure 4 ijms-23-09963-f004:**
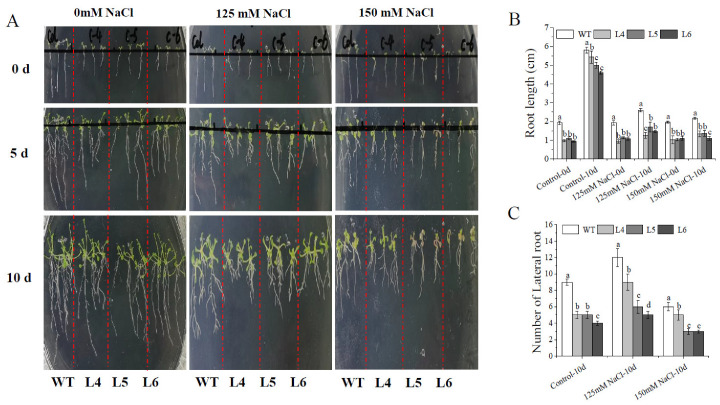
Phenotypes of the transgenic Arabidopsis and wild-type seedlings under 1/2 MS with different concentrations of NaCl: (**A**) seedlings growing on 1/2 MS with 0 mM, 125 mM, and 150 mM NaCl, respectively; (**B**) primary root length (cm); (**C**) number of lateral roots. Different letters indicate significant differences among L4, L5, L6, and WT under the same treatment (*p* < 0.05).

**Figure 5 ijms-23-09963-f005:**
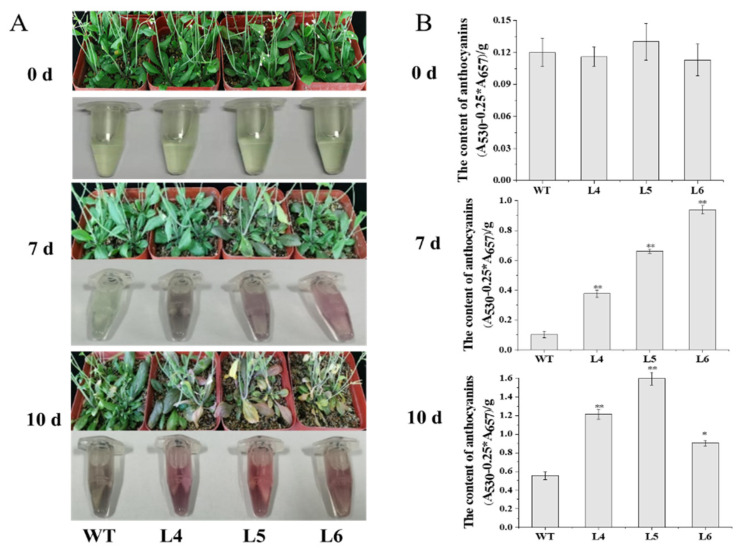
Phenotypes and the anthocyanins content of the transgenic Arabidopsis and wild-type seedlings under irrigation with 200 mM NaCl: (**A**) seedlings irrigated with 200 mM NaCl at 0, 7, and 10 d, respectively; (**B**) anthocyanin content in rosette leaves at 0, 7, and 10 d, respectively. **, * indicate the significant difference among L4, L5, L6, and WT at *p* < 0.01 and *p* < 0.05, respectively.

**Figure 6 ijms-23-09963-f006:**
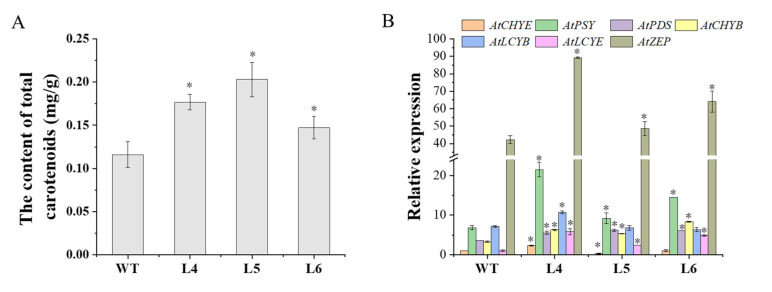
Total carotenoid content and transcript levels of the carotenoid biosynthesis genes in the rosette leaves of transgenic Arabidopsis and WT plants under irrigation with 200 mM NaCl for 7 days: (**A**) total carotenoids content, (**B**) relative expression of carotenoid biosynthesis genes. *AtActin* was used as an internal control. Expression of *AtCHYE* in WT was used as the calibration. Significant difference in gene expression between WT and transgenic lines (L4, L5, and L6) at * *p* < 0.05 by Student’s *t*-test.

**Figure 7 ijms-23-09963-f007:**
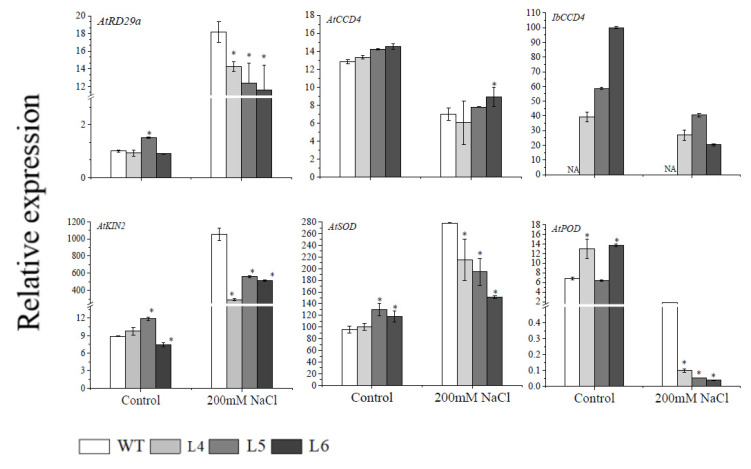
Transcript levels of the resistance-related genes in the transgenic Arabidopsis and WT plants under irrigation with 200 mM NaCl for 7 days. *AtActin* was used as an internal control. The expression of *AtRD29a* in WT was used as the calibration. Significant difference between L4, L5, and L6 and WT at * *p* < 0.05 by Student’s *t* -test.

**Figure 8 ijms-23-09963-f008:**
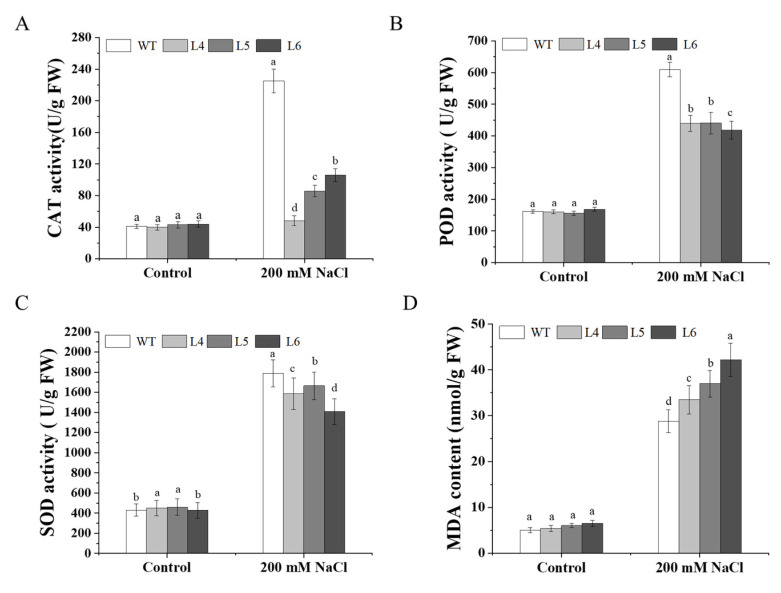
SOD, POD, and CAT activity and MDA content in transgenic Arabidopsis and WT plants under irrigation with 200 mM NaCl for 7 days: (**A**) CAT activity (U/g FW), (**B**) POD activity (U/g FW), (**C**) SOD activity (U/g FW), (**D**) MDA content. Data presented as mean ± SE (*n* = 3). Different letters indicate significant differences among L4, L5, L6, and WT at *p* < 0.05.

## Data Availability

The data presented in this study are available on request from the corresponding author.

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
