# Peer review of "Overexpression of Sweet Potato Carotenoid Cleavage Dioxygenase 4 (IbCCD4) Decreased Salt Tolerance in Arabidopsis thaliana"

_ijms, 2022, doi:10.3390/ijms23179963_

Round 1
Reviewer 1 Report
The article entitled “Overexpression of Sweetpotato Carotenoid Cleavage Dioxygenase 4 (IbCCD4) Decreased Salt Tolerance in Arabidopsis thaliana” reported ectopic expression of IbCCD4 affect salt stress sensitivity in Arabidopsis. Previously, CsCCD4 (from Camellia sinensis) overexpression, the transgenic Arabidopsis showed increased salt stress tolerance, but IbCCD4 expression showed the opposite result. Transgenic plants were more sensitive to salinity stress due to reduced expression of antioxidant genes. In addition, the accumulation of anthocyanins was increased by salt stress in transgenic Arabidopsis. The authors conclude that the increased salinity sensitivity of transgenic plants may be due to a possible malfunction of IbCCD4 on an unknown cleavage target Despite the contrary findings of previous reports, this article deserves publication in the IJMS on other cases of CCD research. However, there are several questions for peer review.
1. Root elongation of IbCCD4 overexpressed Arabidopsis plants were shorter than WT under normal condition even in 10 day after measurement. Therefore, the different root length of samples may not due to NaCl stress. Please discuss this question.
2. Root elongation of IbCCD4-overexpressing Arabidopsis plants was shorter than that of WT under normal conditions even at 10-day measurements. Therefore, the different root lengths in the sample may not be due to NaCl stress. Discuss this question.
3. The authors conclude that the decrease in salt tolerance is due to an unexpected function of IbCCD4 in Arabidopsis. If so, how different is the sequence of IbCCD4? What motifs are different from homologous genes such as CsCCD4, MdCCF4c and OfCCD4? If possible, compare corresponding protein motifs for further study.
Reviewer 2 Report
The paper mainly showed the expression of sweetpotato 1IbCCD4 was significantly induced under salt and drought stresses, and overexpression of IbCCD4 reduced the salt tolerance of transgenic Arabidopsis. Some of questions and revision points are rest as follows.
*Introduction:
-Lines 51-52: A previous reports would be needed about the studies of overexpression of CCD4, if possible.
*Results:
-Line 103: The title of 2.3 would be contained ‘Transgenic’ like ~in Transgenic Arabidopsis. And the same check point at line 139, the title of 2.4.
-Line 132: Fig. 3B is needed marks of 0d, 7d, 10d as shown on plate of Fig. 3A.
*Discussion
-Lines 205-214: A previous studies would be needed about the studies of a biotic stress and purple color, if possible, at the end of the line 214.
*Other minor revision was checked on the manuscript paper.

Reviewer 3 Report
- At the introduction, I think it is necessary to write a few ideas related to the importance of carotenoids for human health.
- Also in the introductory part, it would be interesting for the reader to know if there are sweet potato varieties obtained through classical breeding, with resistance to salt stress. This, in the context where the long-term risks of genetically modified organisms are not known.
- In order for the introductory table to be more complete, it would be useful to present in a few figures the situation of saline soils in the world and in China.
Otherwise, the paper is very logically structured and presented, and I congratulate the authors for this.
